# Prevalence of Hypertension and Obesity: Profile of Mitochondrial Function and Markers of Inflammation and Oxidative Stress

**DOI:** 10.3390/antiox12010165

**Published:** 2023-01-10

**Authors:** Andrés García-Sánchez, Luis Gómez-Hermosillo, Jorge Casillas-Moreno, Fermín Pacheco-Moisés, Tannia Isabel Campos-Bayardo, Daniel Román-Rojas, Alejandra Guillermina Miranda-Díaz

**Affiliations:** 1Department of Physiology, University Health Sciences Center, University of Guadalajara, Guadalajara 44430, Jalisco, Mexico; 2Department of Chemistry, University Center for Exact Sciences and Engineering, University of Guadalajara, Guadalajara 44430, Jalisco, Mexico

**Keywords:** ATP synthase, mitochondrial function, carbonyl groups, oxidative stress, obesity

## Abstract

Obesity and hypertension are health problems of increasing prevalence in developed countries. The link between obesity and hypertension is not yet fully determined. Oxidative stress (OS) and mitochondrial function may play a role in obesity-associated hypertension. A cross-sectional study with 175 subjects with normal weight, overweight, or obese who attended a medical check-up was included. The subjects were divided according to the body mass index (BMI) into normal-weight (n-53), overweight (n-84), and obesity (n-38). Hypertension was also evaluated. To measure mitochondrial function, ATP hydrolysis and ATP synthesis in platelets and serum, respectively, were determined. Superoxide dismutase (SOD), catalase, lipohydroperoxides, 8-isoprostanes, carbonyl groups in proteins, nitric oxide (NO) metabolites, 8-hydroxy-2′-deoxyguanosine (8-OHG), 8-oxoguanine glycosylase (hOGG1), tumor necrosis factor-alpha (TNF-α) and interleukin 6 (IL-6) were measured by standard colorimetric or immunoassay methods. Obese subjects showed lower ATP hydrolysis activity than normal weight and overweight subjects (*p* < 0.01). No differences between those groups were found in ATP synthase and catalase activities, lipid hydroperoxides, carbonyl groups in proteins, 8-isoprostanes, and NO metabolites. In the obesity group, SOD activity *(p* < 0.01) was decreased while 8-OHG *(p* < 0.01) was increased. Subjects with hypertension showed increased 8-OHG (*p* < 0.01) and less reparative enzyme (hOGG1 *p* = 0.04) than subjects with normal weight. Moreover, we found a decrease of SOD *(p* < 0.01), catalase activities (*p* = 0.04), NO metabolites (*p* < 0.01), and increases of carbonyl groups in proteins (*p* = 0.01), TNF-α (*p* < 0.01) and IL-6 (*p* < 0.01 in hypertensive subjects. Obese subjects show a decrease in ATP hydrolysis. The decrease in ATP hydrolysis rate and ATP synthesis and an increase in OS and inflammation markers were associated with the hypertensive state.

## 1. Introduction

Obesity is a critical health problem that affects about 36.9% of men and 38.0% of women worldwide [1]. Being overweight or having obesity is a risk factor for diabetes mellitus (DM), cardiovascular disease (CVD), and cancer [2]. Hypertension is the primary preventable risk factor for cardiovascular disease, and obesity-related hypertension is the major hypertension type [3]. There is a relationship between the increased prevalence of hypertension and obesity, but the underlying mechanisms of obesity-related hypertension remain under investigation [4]. Obesity is a systemic inflammation that promotes the immune system’s activation, leading to a pro-inflammatory state and oxidative stress (OS) [5,6]. OS and inflammation are key mechanisms of endothelial dysfunction and arterial damage, linking these risk factors to vascular disease and arterial stiffness [7]. Specific oxidative pathways involving pro-oxidant enzymes appear to play an important role in producing reactive oxygen species (ROS) [8,9]. Increased ROS generation may damage mitochondrial proteins and DNA leading to decreased bioenergetics and mitochondrial efficiency [10]. Damage to mitochondrial membrane permeability impairs electron transport through increased ROS and the ATP-synthesis machinery’s dysfunction [11]. Excess nutrient intake, physical inactivity, and weight gain can increase ROS production, favoring mitochondrial dysfunction [12]. Experimental and clinical studies have shown that mitochondrial dysfunction may participate in the mechanisms of hypertension [13]. In consonance, a cross-sectional study showed that damage to mitochondrial DNA is a risk factor for DM, atherosclerosis, and hypertension [14].

Mitochondrial ATP synthase is a multi-subunit-enzyme complex that catalyzes ATP synthesis using a transmembrane electrochemical proton gradient generated by the electron transport chain [15]. The enzyme can function as ATP synthase (ATP synthesis) or ATPase (ATP hydrolysis). The enzyme is composed of two main portions: F_O_, a sector portion embedded in the inner mitochondrial membrane (IMM), and F_1_, a soluble catalytic sector bound through two stalks to F_O_ [16]. It has been suggested that the F_1_ portion could be an important site for oxidative and nitrosative modifications, particularly the β-subunits [17,18]. Oxidative and nitro-oxidative modifications of a specific cysteine residue in ATP synthase modulate its enzymatic activity in an animal model of heart failure [19]. Obese individuals have reported decreased ATP synthesis rates in skeletal muscle [20]. However, the clinical information about the possible association between mitochondrial dysfunction and the early complications related to being overweight and obese is unclear.

The study aimed to evaluate the prevalence of mitochondrial ATP synthase activity, inflammatory cytokines, and OS markers in subjects who were overweight or obese with or without hypertension.

## 2. Materials and Methods

An analytical cross-sectional study was carried out on subjects with normal body weight, overweight, or obese with or without hypertension who attended a medical check-up. It was recorded as height, weight, and body mass index (BMI). BMI was reported in kg/m^2^. The criteria of the World Health Organization (WHO) for BMI were used: normal weight (18.5–24.9 kg/m^2^), overweight (25–29.9 kg/m^2^), and obesity (≥30 kg/m^2^) [21]. Hypertension was recorded as systolic blood pressure ≥130 mmHg or diastolic blood pressure >80 mmHg [22]. Type 2 DM was established as fasting plasma glucose ≥126 mg/dL [23].

Dyslipidemia criteria included high LDL cholesterol (≥160 mg/dL), low HDL cholesterol (<40 mg/dL), or high triglycerides (≥150 mg/dL) [23]. No maintenance therapy was excluded.

When interviewing to take the medical history of the subjects, some mentioned taking proton pump blockers for gastritis or analgesics for headaches. Those drugs did not change.

Subjects who reported kidney disease, DM, cerebrovascular disease, hepatitis, or taking antioxidants during the last three months were not included. Samples of 8 h fasting blood were taken, 5 mL with ethylenediaminetetraacetic acid (EDTA), and 5 mL in the dry tube. The samples were centrifuged at 3000 rpm for 10 min at room temperature to obtain plasma, erythrocytes, and serum. Blood fractions were stored at −80 °C until final processing.

### 2.1. Sub-Mitochondrial Membranes of Platelets

Sub-mitochondrial membranes of platelets were obtained by centrifuging blood samples at 2980× *g* for 15 min at 4 °C. The supernatant was removed, and 200 µL cold buffer was added (composition [in mmol/L]: NaCl 140, KCl 4.7, MgCl_2_ 1.2, KH_2_PO_4_ 1.2, dextrose 11, HEPES 15) to the pellet and the samples homogenized. An aliquot of 70 µL of samples containing platelets was stored at −80 °C until the final processing.

### 2.2. ATP Synthesis Activity

ATP synthesis rate was assayed using a coupled-enzyme assay at 37 °C. Briefly, 0.5 mL of the reaction mixture (125 mM KCl, 25 mM sucrose, 40 mM Hepes (pH 7.5), 5 mM MgCl_2_, 0.1 mM ethylene glycol-bis tetra-acetic acid (EGTA), 1 mM NADH, 20 mM glucose, 46 units hexokinase, 2 mM Mg^2+^-ADP and 3 mM of inorganic phosphate) was preincubated at 37 °C for 2 min. Then, 60 µL of platelet sample was added. The reaction was followed for 10 min. Then, it was quenched with 50 µL of a mixture containing 25 mM EDTA, 2 µM carbonyl cyanide m-chlorophenyl hydrazone, and 25 µg oligomycin. The reaction mixture was boiled for 10 min and centrifuged at 5000× *g* for 5 min. Then 50 µL of a mixture of NADP (0.5 mM) and glucose-6- phosphate dehydrogenase (30 units) were added to the supernatant. The absorbance of the samples was recorded at 340 nm.

### 2.3. ATPase Activity

The reaction medium (0.5 mL) for ATPase determination contained 125 mM KCl, 40 mM Hepes-KOH (pH 8.0), 0.1 mM EGTA, and 3 mM MgCl_2_. The reaction was initiated by adding 60 µL of platelet samples and 20 µL of 100 mM ATP. After 10 min, the reaction was quenched with 50 µL of 30% (*w/v*) trichloroacetic acid. Afterward, the sample was centrifuged for 10 min at 3500 rpm, and 600 μL of the supernatant was recovered, and 300 µL of 3.3% ammonium molybdate was added, followed by 100 μL of 10% ferrous sulfate. The absorbance of the samples was recorded at 660 nm. These data were referred to as a phosphate curve.

### 2.4. Antioxidants

#### 2.4.1. Superoxide Dismutase Activity

The kit (SOD No. 706002, Cayman Chemical Company^®^, Ann Arbor, MI, USA) was used according to the manufacturer’s instructions. The serum samples were diluted 1:2 in a buffer before the colorimetric assay. Then, a radical detector was used with xanthine oxidase to produce a color signal. The color development was read at a wavelength of 440 nm. The dilution factor was used to calculate the results.

#### 2.4.2. Catalase Activity

Thirty microliters of erythrocytes were added to a microtube with 300 µL of distilled water. Subsequently, 500 µL of hydrogen peroxide solution in sodium-potassium phosphate buffer, pH 7.4, was added. The reaction was followed for 3 min and stopped with 100 µL of ammonium molybdate solution. Then, 20 µL was added to a microplate with 180 µL of water, and the absorbance of the samples was measured at 379 nm. The activity was reported as KU/mL.

### 2.5. Oxidants

#### 2.5.1. Lipid Hydroperoxides

Lipid hydroperoxides (LHP) quantification was performed as described previously [23] with slight modifications. Before starting the analysis, 200 µL of erythrocytes were lysed with 1.4 mL of distilled water and washed three times. To the precipitate from the last wash, we added 600 µL of reagent solution (containing xylenol orange, ferrous ammonium sulfate, methanol, butylated hydroxytoluene, and H_2_SO_4_). The sample was incubated for 10 min at room temperature. Subsequently, the samples were centrifuged for 10 min at 14,000× *g*, and the absorbance of the supernatant was read at 560 nm.

#### 2.5.2. Carbonyl Groups in Proteins

Plasma (200 μL) was mixed with 500 μL of 10 mM 2, 4-dinitrophenylhydrazine in 2 M HCl and incubated for 1 h at room temperature. After that, 333 μL of trichloroacetic acid (30%, *p*/*v*) was added, followed by centrifugation at 14,000× *g* for 20 min. The resulting pellet was washed three times with 1 mL of ethanol-ethyl acetate solution (1:1, *v*:*v*). Then 600 μL of guanidine hydrochloride 6 M was added to the final pellet, followed by incubation for 15 min at room temperature. The absorbance of the samples was read at 370 nm.

#### 2.5.3. 8-Isoprostane

Plasma samples were analyzed according to the competitive ELISA assay (ABCAM ab175819^®^, Cambridge, United Kingdom). 100 µL of sample and horseradish peroxidase (HRP) conjugate were plated on a plate pre-coated with capture antibodies to 8-isoprostanes. A TMB (3,3′,5,5′-Tetramethylbenzidine) substrate was used for the colorimetric reaction. The reaction was stopped with 2N sulfuric acid. Absorbance was read at 450 nm.

#### 2.5.4. Nitric Oxide Metabolites

Before the assay, plasma samples were deproteinized by adding zinc sulfate (6 mg of zinc sulfate was added to 400 μL of the sample), vortexed for one min, and centrifuged at 10,000× *g* for 10 min at 4 °C. The kit NB98, Oxford Biochemical^®^, Oxford, MI, USA was used to measure NO metabolites. Absorbance was read at 540 nm.

#### 2.5.5. Markers of Oxidative Damage and Repair of DNA

##### 8-Hydroxy-2′-Deoxyguanosine (8-OHG)

The assay was performed with the reagents kit 8-OHG (No. ab201734 Abcam^®^, Cambridge, United Kingdom). The competitive ELISA assay was performed with 50 µL of a serum sample, and the color signal was measured at 450 nm. The duplicate standard intra-assay CV was 6.9%.

##### 8-Oxoguanine-DNA-N-Glycosylase-1

We followed ELISA kit hOGG-1 MBS702793 (My BioSource^®^, San Diego, CA, USA) which consists of a sandwich-type ELISA. The sample was placed in a 96-well microplate pre-coated with an antibody specific for hOGG1. A biotin detection antibody followed by an avidin-HRP conjugate was used. TMB was used as a substrate, and absorbance was measured at 450 nm with the correction wavelength set as 540 nm.

### 2.6. Pro-Inflammatory Cytokines

#### TNF-α and IL6

ELISA kits 900-K25 and 900-K16 (Peprotech, Rocky Hill, NJ 08553, USA^®^) were used to determine the TNF-α and IL-6, respectively. Both cytokines had a detection limit of 32 pg/mL. The cytokines quantification was made with 100 μL of a plasma sample. The plate was read at a wavelength of 405 nm with a correction set at 650 nm using a Synergy HT multi-detection microplate reader (Bio-Tek, Winooski, VT, USA).

### 2.7. Statistical Analysis

Data were expressed as median (25–75 percentiles) or percentages. The Kolmogorov–Smirnov test was used to analyze the normality distribution of data. We used the Kruskal–Wallis test with the post-hoc Dunn-Bonferroni test for pairwise comparison to analyze the differences between multiple groups for continuous variables. The difference between the groups was calculated with the Chi^2^ test for categorical variables. Two-tailed *p* ≤ 0.05 was considered statistically significant. SPSS for Windows version 20.0 (IBM SPSS statics Inc., Chicago, IL, USA) was used.

## 3. Results

Sixty-two men aged 41.4 ± 14.8 and 113 women aged 50.3 ± 12.6 were included. The prevalence of subjects with dyslipidemia was similar in normal-weight, overweight, and obesity. Hypertension was more prevalent in overweight and obese subjects than in normal-weight subjects (*p* < 0.01). (Table 1).

The ATP hydrolysis activity was less in obese subjects *p* < 0.01, 9.3 nmol/min/mg protein (7.5–15.6 nmol/min mg protein), compared to normal-weight subjects, 15.9 nmol/min mg protein (12.1–18.8 nmol/min mg protein) and overweight, 15.3 nmol/min mg protein (10.1–20.5 nmol/min mg protein). Obese subjects had a higher concentration of the oxidative DNA damage marker (8-OHG), 23.0 ng/mL (3.7–70.9 ng/mL), compared to normal-weight, 4.8 (1.1–39.4 ng/mL), and overweight subjects 1.9 (1.0–23.8), *p* < 0.01. SOD activity was lower *p* < 0.01 in obese subjects 0.4 U/mL (0.1–0.6 U/mL) compared to subjects with normal-weight 0.7 U/mL (0.5–1.0 U/mL) and overweight 0.6 U/mL (0.3–1.1 U/mL). (Table 1).

The results were compared between subjects with and without hypertension without considering their body weight with the following results.

### 3.1. Mitochondrial Function in Hypertension

The ATP hydrolysis was significantly higher in subjects without hypertension, 15.9 nmol/min mg protein (11.4–20.4 nmol/min mg protein), *p* < 0.01 than in hypertensive subjects, 9.0 nmol/min mg protein (7.6–12.9 nmol/min mg protein). The ATP synthesis was similar between subjects without hypertension, 127.27 (96.70–144.71 nmol/min mg protein) those with hypertension, 108.08 (88.19–145.43 nmol/min mg protein). (Table 2).

### 3.2. Proinflammatory Cytokines in Hypertension

The levels of the pro-inflammatory cytokine TNF-α were increased in hypertensive subjects, 516.1 pg/mL (279.7–856.3 pg/mL), *p* < 0.01 compared to subjects without hypertension, 197.7 pg/mL (142.0–197.7 pg/mL). The IL-6 behaved similarly to TNF-α with a significant increase in subjects with hypertension, 235.3 pg/mL (83.1–482.4 pg/mL), *p* < 0.01 compared to subjects without hypertension, 64.9 pg/mL (50.6–247.9 pg/mL). (Table 2).

### 3.3. DNA Oxidative Damage Markers in Hypertension

The oxidative DNA damage marker (8-OHdG) was found to be significantly increased in hypertensive subjects, 45.1 ng/mL (5.8–70.9 ng/mL), *p* < 0.01 compared to subjects without hypertension, 2.5 ng/mL (0.9–22.8 ng/mL). The DNA repair enzyme was significantly decreased in hypertensive subjects, 0.04 ng/mL (0.02–3.6 ng/mL), *p =* 0.04 front non-hypertensive subjects, 0.6 ng/mL (0.2–4.2 ng/mL). (Table 2).

### 3.4. Antioxidants in Hypertension

The activity of the enzyme catalase in hypertensive subjects showed a significant decrease, 119.8 KU/mL (91.1–167.1 KU/mL), *p =* 0.04 compared to the enzyme’s activity in non-hypertensive subjects, 149.9 KU/mL (126.4–171.6 KU/mL). SOD enzyme activity was found to be significantly decreased in subjects with hypertension, 0.36 U/mL (0.13–0.62 U/mL), *p* < 0.01 compared to enzyme activity in non-hypertensive subjects, 0.7 U/mL (0.4–1.0 U/mL). (Table 2).

### 3.5. Oxidants in Hypertension

Carbonyl groups in proteins were significantly increased in subjects with hypertension, 4.0 µmol (3.3–5.5 µmol), *p* = 0.01 compared to non-hypertensive subjects, 3.1 µmol (1.9–4.5 µmol). 8-IP levels were higher in non-hypertensive subjects *p* = 0.04, 31.6 pg/mL (20.5–66.5 pg/mL), *p* = 0.04 than in hypertensive subjects, 23.9 pg/mL (17.7–35.5 pg/mL). NO levels in non-hypertensive subjects were higher *p* < 0.01, 300.0 µM (253.0–376.1 µM) than in hypertensive subjects. 251.7 µM (227.6–290.1 µM). The levels of lipid hydroperoxides were similar in subjects with and without hypertension. (Table 2).

### 3.6. Mitochondrial Function in Overweight or Obese Subjects with and without Hypertension

Table 3 shows the results of OS markers in overweight or obese subjects with and without hypertension. ATP hydrolysis was significantly lower in hypertensive subjects with 9.4 nmol/min mg protein (8.3–12.9 nmol/min mg protein), *p* = 0.01 front overweight subjects without hypertension, 16.2 nmol/min mg protein (12.5–20.8 nmol/min mg protein). The ATP synthesis was similar in overweight, obese subjects with and without hypertension.

### 3.7. DNA Oxidative Damage Markers in Overweight, Obesity, and Hypertension

The significant overexpression of the marker for oxidative damage to DNA (8-OHG) stands out in subjects with overweight and hypertension, 40.2 ng/mL (2.3–72.2 ng/mL), *p* = 0.01 compared to the levels obtained in overweight subjects without hypertension, 1.7 ng/mL (0.4–16.1 ng/mL). Inversely, levels of the DNA repair enzyme (hOGG1) in overweight hypertensive subjects were significantly lower, 0.02 ng/mL (0.01–0.6 ng/mL), *p* = 0.01 than overweight subjects without hypertension 0.5 ng/mL (0.3–3.0 ng/mL).

### 3.8. Oxidants in Overweight, Obesity, and Hypertension

The level of carbonyl groups in proteins was increased in subjects with hypertension and overweight, 4.2 µmol (3.0–5.8 µmol), *p* = 0.5. Levels of carbonyl groups in proteins were lower in overweight subjects without hypertension, 3.0 µmol (1.6–4.5 µmol). 8-IP levels were lower in subjects with overweight and hypertension, 19.2 pg/mL (17.2–26.0 pg/mL), *p* = 0.01, than in overweight subjects without hypertension, 36.4 pg/mL (20.2–69.2 pg/mL). A significant decrease in NO metabolite levels was found in subjects with overweight and hypertension, 249.6 µM (237.3–269.8 µM), *p =* 0.01, compared to the levels obtained in subjects with overweight without hypertension 323.7 µM (259.9–383.2 µM).

### 3.9. Antioxidants in Overweight, Obesity, and Hypertension

In subjects with obesity and hypertension, the activity of the antioxidant enzyme catalase was significantly decreased, 112.8 KU/mL (95.1–140.6 KU/mL), *p* = 0.02, contrary to what was found in subjects with obesity without arterial hypertension, 142.4 KU/mL (114.8–177.0 KU/mL). SOD enzyme activity was found significantly decreased in overweight and hypertensive subjects, 0.2 U/mL (0.1–0.6 U/mL), *p* < 0.01, contrary to that obtained in overweight subjects without hypertension, where the enzyme activity was increased, 0.8 U/mL (0.4–1.1 U/mL).

## 4. Discussion

Obesity is a growing health problem in many developed countries [24]. In countries like Mexico, the prevalence of obesity and overweight has been estimated at 62% [25]. This study shows a prevalence of overweight and obesity of 66%, being more prevalent in women. However, this prevalence is expected to increase to 88% in the next 40 years [26]. Obesity is associated as a risk factor for 18 health problems that threaten people’s lives, such as; some types of cancer, heart attack, respiratory problems, type 2 DM, and hypertension [27]. Among the comorbidities of overweight and obesity, hypertension is one of the most common [28]. In this study, obese subjects had higher hypertension prevalence than normal-weight subjects. Hypertension in obesity was more prevalent in women than in men. The literature indicates that the prevalence of hypertension due to obesity can represent about 40% of total hypertension [29].

The subjects who participated in this study were volunteers who were not known to be ill. The subjects attended the check-up as healthy subjects without pathologies. In this sense, it was expected that the prevalence of hypertension would be different from that of the general population [30]. Hypertension displayed by the study subjects can be considered undiagnosed or untreated hypertension. The present study shows a prevalence of undiagnosed hypertension of 18%, which is similar to that reported by other publications that contemplate a prevalence of approximately 13% [31].

The relationship between obesity and hypertension is not entirely clear. Some authors point out that obesity works as a hypertensinogenic that increases the risk of raising blood pressure in susceptible subjects [32]. The mechanisms that intervene in hypertension in obesity include inflammatory processes, chronic diseases, and OS [33]. This study evaluated the inflammatory profile, OS markers, and mitochondrial function in subjects who did not know they were carriers of pathological conditions such as overweight, obesity, and hypertension.

One of the characteristics evaluated in the present study was markers of mitochondrial function. Measurement of ATPase hydrolytic activity and ATP synthesis are used as indicators of mitochondrial function [34]. The energy stored in the phosphoanhydride bond of ATP drives many metabolic processes, including muscle contraction, cell motility, and DNA synthesis, among many others [35]. ATP turnover is also remarkable: the human body uses ATP daily, and cells must continually regenerate ATP from the products of its hydrolysis (ADP and phosphate) to keep up [36]. Excessive intake of high-calorie nutrients produces high levels of free fatty acids and ROS, leading to mitochondria dysfunction and may alter the functioning of mitochondrial ATP synthase [37,38]. However, the results in the present document showed no differences in systemic blood ATP synthesis rate between subjects with normal weight, overweight, and obesity. Previous studies show that subjects with obesity have decreased ATP synthesis in muscle [39]. However, studies evaluating the systemic measurement of systemic ATP synthesis in obesity are lacking.

On the other hand, ATP hydrolysis was lower in obese subjects compared to normal-weight and overweight subjects. This study indicates that obesity may be related to decrease systemic ATP hydrolysis. Mitochondrial dysfunction can be caused by early metabolic disorders or genetic factors and is implicated in the development of insulin resistance [40,41,42]. Subjects with hypertension had lower ATP hydrolysis activity than subjects without hypertension, significantly for overweight subjects. Rodrigo R et al. describe that subjects with hypertension present decreased ATPase activity, probably mediated by ROS increase [43]. Moreover, inflammation contributes to the pathogenesis of hypertension.

Biomarkers of inflammation, including various cytokines and products of the complement pathway, are elevated in humans with hypertension. Emerging evidence suggests that hypertension is accompanied and initiated by the activation of complement, the inflammasome, and a change in the phenotype of circulating immune cells, particularly myeloid cells. High-dimensional transcriptomic analyzes provide insight into new subclasses of immune cells that are likely to be detrimental to hypertension. These inflammatory events are interdependent, and the adaptive immune system compromises through mechanisms that involve OS, modification of endogenous proteins, and alterations in the processing and presentation of antigens [44]. The association between hypertension and inflammation has been demonstrated; however, it is not clear whether inflammation is predominantly a cause or an effect of hypertension [45].

In this study, the DNA oxidative damage marker (8-OHG) overexpression in subjects with hypertension was notable. Even the overexpression of this marker predominated in overweight and hypertensive subjects than in obese and hypertensive subjects.

Oxidative damage to DNA has mutagenic potential. Previous studies described the high concentration of oxidative DNA damage related to obesity and tumorigenesis [46]. Increased levels of 8-OHG are related to cardiovascular diseases [47]. Oxidative damage to DNA can form mutations that cause endothelial dysfunction, including impaired vaso-regulation, impaired barrier function, and the adhesion of molecules to the endothelium [48]. Previous studies indicate increased concentrations of 8-OHG in subjects with hypertension compared to normotensive subjects [49]. Therefore, it is recommended to maintain body weight within normal standards to prevent conditions associated with oxidative injury and associated factors such as arterial hypertension observed in the present study.

Conversely, the behavior of the DNA repair enzyme (hOGG1) was significantly downregulated in subjects with hypertension and overweight. Even an enzyme decrease was found in obese subjects (Table 3). The hOGG1 is the critical component responsible for removing oxidative damage to DNA [50]. The DNA repair system is essential to overcome oxidative damage and maintain the integrity of the DNA structure. ROS induction in human cells has been reported to actively inhibit DNA repair by increasing protein oxidation [51]. In a study published in 2020, the authors report slight DNA-repairing enzyme levels in overweight and obese subjects with a significant decrease in the enzyme in subjects with abdominal visceral fat [52]. This data suggests that being overweight and having high blood pressure could indicate an imbalance of oxidative damage to DNA.

Carbonyl groups in proteins are markers of oxidative damage to proteins. These markers probably participate in the pathogenesis of several diseases, including hypertension [53]. The carbonyl groups in proteins are a severe marker of the operating system, capable of irreversibly damaging the structure of the protein and causing the inhibition of enzymatic activity with greater susceptibility to proteolysis. There is little scientific information available on the oxidation of proteins in metabolic syndrome. However, high levels of carbonyl groups in proteins are often detected in subjects with a metabolic syndrome characterized by DM, dyslipidemia, obesity, and hypertension. Obesity, insulin resistance, and DM are often associated with increased protein carboxylation [54]. In the present study, the protein carboxylation marker was found to significantly increase mainly in subjects with overweight and hypertension, which means that hypertension could play an important role in the structural damage of overweight and obesity proteins.

Isoprostanes are chemically stable products of the peroxidation of lipids derived from arachidonic acid, and its quantification provides a new approach to evaluating the OS in vivo. Therefore, the levels of isoprostanes increase during the OS. In consonance, the plasma level of 8-Isoprostane was found to be elevated in elderly subjects with hypertension [55]. However, the isoprostane levels were found to be significantly lower in hypertension than those without hypertension, and this trend was observed in overweight subjects.

This study’s significant decrease in NO levels in obese hypertensive subjects is striking. The link between the two factors, hypertension and overweight, amply justifies the decrease in NO levels according to recent reports where hypertension is a significant risk factor for cardiovascular disease [56]. Endothelial dysfunction, characterized by impaired NO bioavailability, is a considerable risk factor for hypertension and cardiovascular disease and may represent an essential link between the conditions. Evidence suggests that NO plays a critical role in blood pressure regulation and that altered NO bioactivity is a crucial component of hypertension [57]. Hypertension is a common sequela of obesity, and it is estimated that obesity contributes to 75% of cases of hypertension in men and 65% in women. Obese subjects are often resistant to standard antihypertensive medication. Understanding the precise mechanisms underlying the link between obesity and hypertension has presented obstacles to developing new and effective therapy. However, obesity was recently associated with reduced bioavailability of NO, apparently caused by impaired transport of the substrate L-arginine. There is clear evidence that endothelial dysfunction plays an essential role in the pathogenesis of hypertension, including the association with overweight and obesity [58].

SOD is the main antioxidant defense system against superoxide anion (O_2_^−^) consisting of three SOD isoforms: cytoplasmic Cu/ZnSOD (SOD1), mitochondrial MnSOD (SOD2), and extracellular Cu/ZnSOD (SOD3), all enzymes require metal (Cu, Zn or Mn) for activation. Excess O_2_^−^ anion plays an important role in the pathogenesis of atherosclerosis and hypertension. SOD catalyzes the conversion of O_2_^−^ to H_2_O_2_ for its participation in cell signaling. SOD plays a fundamental role in the inhibition of the oxidative inactivation of NO, which prevents the formation of peroxynitrites and endothelial and mitochondrial dysfunction [59]. In the present study, SOD enzyme activity was significantly decreased in obese subjects. SOD enzyme activity was even found to be decreased in hypertensive and overweight subjects, which could suggest a link between increased body weight, hypertension, and decreased SOD activity.

Catalase enzyme activity was decreased in obese and overall hypertensive patients. Catalase is a crucial antioxidant enzyme that mitigates OS to a great extent by destroying cellular H_2_O_2_ to produce oxygen and water. Catalase deficiency or malfunction is related to the pathogenesis of many age-associated degenerative diseases, such as DM, Alzheimer’s disease, Parkinson’s disease, and hypertension. Adipose tissue is an organ that secretes adipokines and generates ROS. Adipose tissue is considered an independent factor for the genera of the systemic OS. There are some mechanisms by which obesity produces OS; the first is mitochondrial and peroxisomal oxidation of fatty acids, which can produce ROS in oxidation reactions. Another mechanism is excessive oxygen consumption, which generates free radicals in the mitochondrial respiratory chain that is coupled to oxidation by phosphorylation in the mitochondria. High-fat diets are capable of generating ROS by altering oxygen metabolism [60]. When the fatty tissue increases, it is possible to find a decreased activity of the antioxidant enzymes, as in the present study. The overproduction of ROS and the decrease in antioxidant capacity give rise to endothelial dysfunction characterized by a reduction in NO’s bioavailability with an increase in the contractile capacity of the endothelium. These factors favor atherosclerotic disease and hypertension [61].

In conclusion, obese subjects show a decrease in ATP hydrolysis. The catalase activity increases along with ATPase activity in overweight and obese subjects. Decreased SOD, hOGG1, and increased 8-OHdG are found in obese subjects. Oxidative stress, pro-inflammatory cytokines, and oxidative DNA damage are greater in subjects with hypertension, while antioxidant defenses, DNA damage repair enzyme, and ATP hydrolysis rate are more significant in subjects without hypertension. Further studies on the implication of these findings in comorbidities may clarify mitochondrial dysfunction in diseases derived from obesity.

Study strengths. This exploratory study shows mitochondrial ATP synthase activity and ATP hydrolysis in normal-weight, overweight and obese subjects. Insufficient information addresses the rate of ATPase activity and ATP synthesis in obesity and hypertension. The findings of the present work may help implement prospective or intervention studies for the prevention or treatment of comorbid obesity related to mitochondrial function.

Limitations of the study. A cross-sectional study cannot determine the causality between alterations in mitochondrial function, OS, and increased body mass. Subjects who spontaneously attended a medical check-up were included, including self-referral or on the recommendation of the care center screening. The onset of obesity is currently unknown.

## Figures and Tables

**Table 1 antioxidants-12-00165-t001:** Demographic data, ATP synthesis rate, ATP hydrolysis rate, oxidative stress, and proinflammatory markers in subjects with normal weight, overweight, and obesity.

	Normal-Wright	Overweight	Obesity	*p*
	n-53	n-84	n-38	
Gender				
Male n (%)	21 (39.6)	30 (35.7)	11 (28.9)	0.58
Female n (%)	32 (60.4)	54 (64.3)	27 (71.1)	
Age years	49 (33.5–59.0)	45 (33.25–53.7)	53 (45.0–60.0) ^a,b^	<0.01 ^ꞙ^
Hypertension n (%)	4 (7.5)	12 (14.3) ^a^	16 (42.1) ^a^	<0.01 ^ꞙ^
Dyslipidemia n (%)	26 (49.1)	37 (44.0)	16 (42.1)	0.18
Mitochondrial function				
ATP synthesis (nmol/min mg protein)	132.1 (101.7–147.3)	116.1 (92.2–147.2)	107.5 (91.0–132.6)	0.12
ATP hydrolysis (nmol/min mg protein)	15.9 (12.1–18.8)	15.3 (10.1–20.5)	9.3 (7.5–15.6) ^a,b^	<0.01 *
Pro-inflammatory cytokines				
TNF-α (pg/mL)	206.4 (142.0–556.9)	195.6 (146.5–529.3)	496.2 (245.9–777.9)	0.08
IL-6 (pg/mL)	63.9 (50.9–290.2)	66.5 (51.1–184.9)	217.0 (56.5–482.4)	0.12
Markers of oxidative damage and repair of DNA				
8-OHdG (ng/mL)	4.8 (1.1–39.4) ^b^	1.9 (1.01–23.8) ^b^	23.0 (3.7–70.9)	<0.01 *
hOGG1 (ng/mL)	0.59 (0.06–2.8)	0.44 (0.05–1.9)	0.05 (0.02–9.0)	0.68
Antioxidants				
Catalase (KU/mL)	150.2 (128.6–177.8)	152.0 (122.7–171.7)	139.2 (104.8–152.8)	0.11
SOD (U/mL)	0.66 (0.5–1.0)	0.62 (0.3–1.1)	0.4 (0.1–0.6)	<0.01 *
Oxidants				
Lipid hydroperoxides _(_µmol)	3.3 (3.2–3.5)	3.3 (3.2–3.4)	3.3 (3.2–3.5)	0.88
Carbonyl groups in proteins (µmol)	3.34 (1.33–4.2)	3.07 (1.9–4.7)	3.6 (2.8–4.9)	0.13
8-Isoprostane (pg/mL)	31.4 (20.5–62.1)	28.6 (19.1–66.4)	25.5 (18.0–44.0)	0.71
Nitric oxide metabolites (µM)	290.0 (238.2–370.0)	300.5 (249.9–379.2)	265.7 (234.8–327.0)	0.08

Values are median (percentile 25–75), 8-OHdG = 8-Hydroxy-2′-deoxyguanosine, hOGG1 = 8-Oxoguanine glycosylase, SOD = Superoxide dismutase, TNF-α = Tumor necrosis factor-alpha, IL-6= Interleukin 6. a, vs. normal-weight. b, vs. obesity. ^ꞙ^ Chi^2^ test. * Kruskal Wallis test with post-hoc Dunn-Bonferroni test.

**Table 2 antioxidants-12-00165-t002:** Mitochondrial function and oxidative stress markers in hypertensive and non-hypertensive subjects.

	Non-Hypertensive n-143	Hypertensive n-32	*p*
Mitochondrial function			
ATP hydrolysis (nmol/min mg protein)	15.9 (11.4–20.4)	9.0 (7.6–12.9)	<0.01
ATP synthesis (nmol/min mg protein)	127.27 (96.70–144.71)	108.08 (88.19–145.43)	0.24
Pro-inflammatory cytokines			
TNF-α (pg/mL)	197.7 (142.0–197.7)	516.1 (279.7–856.3)	<0.01
IL-6 (pg/mL)	64.9 (50.6–247.9)	235.3 (83.1–482.4)	<0.01
Marker of oxidative damage and repair of DNA			
8-OHdG (ng/mL)	2.5 (0.9–22.8)	45.1 (5.8–70.9)	<0.01
hOGG1 (ng/mL)	0.6 (0.2–4.2)	0.04 (0.02–3.6)	0.04
Antioxidants			
Catalase (KU/mL)	149.90 (126.4–171.6)	119.8 (91.1–167.1)	0.04
SOD (U/mL)	0.7 (0.4–1.0)	0.4 (0.1–0.6)	<0.01
Oxidants			
Lipid hydroperoxides (µmol)	3.32 (3.22–3.44)	3.38 (3.24–3.65)	0.17
Carbonyl groups in proteins (µmol)	3.1 (1.9–4.5)	4.0 (3.3–5.5)	0.01
8-Isoprostane (pg/mL)	31.6 (20.5–66.5)	23.9 (17.7–35.5)	0.04
Nitric oxide metabolites (µM)	300.0 (253.0–376.1)	251.7 (227.6–290.1)	<0.01

Values are median (percentile 25–75), 8-OHdG = 8-Hydroxy-2′-deoxyguanosine, hOGG1 = 8-Oxoguanine glycosylase, SOD = Superoxide dismutase, TNF-α = Tumor necrosis factor-alpha, IL-6 = Interleukin 6. Significance obtained with Mann-Whitney U test, *p* < 0.05.

**Table 3 antioxidants-12-00165-t003:** Mitochondrial function, oxidative stress markers in hypertensive and non-hypertensive subjects with overweight or obesity.

	Overweight			Obesity			
	Non-Hypertensive n-72	Hypertensive n-12	*p* *U-MW*	Non-Hypertensive n-22	Hypertensive n-16	*p* *U-MW*	** p* *U-MW*
Mitochondrial function							
ATP synthesis (nmol/min mg protein)	130.4 (94.0–147.5)	103.6 (80.2–143.5)	0.26	108.7 (94.1–134.1)	107.4 (89.1–128.0)	0.68	0.36
ATP hydrolysis (nmol/min mg protein)	16.2 (12.5–20.8)	9.4 (8.3–12.9)	0.01 *	10.2 (7.9–17.8)	8.1 (7.0–9.5)	0.06	0.02 *
Pro-inflammatory cytokines							
TNF-α (pg/mL)	186.8 (142.0–500.6)	438.7 (207.9–977.8)	0.07	410.0 (160.6–820.9)	604.4 (374.7–767.9)	0.27	0.25
IL-6 (pg/mL)	64.6 (49.3–175.8)	164.3 (65.6–778.8)	0.10	141.2 (49.0–371.4)	265.0 (172.3–482.4)	0.10	0.48
Markers of oxidative damage and repair of DNA							
8-OHdG (ng/mL)	1.7 (0.4–16.1)	40.2 (2.3–72.2)	0.01 *	23.0 (1.6–75.1)	27.6 (6.4–70.8)	0.45	0.03 *
hOGG1 (ng/mL)	0.5 (0.3–3.0)	0.02 (0.01–0.6)	0.01 *	0.04 (0.01–14.40)	0.40 (0.04–6.6)	0.69	0.31
Antioxidants							
Catalase (KU/mL)	151.3 (125.3–171.6)	169.4 (79.8–186.1)	0.81	142.4 (114.8–177.0)	112.8 (95.1–140.6)	0.02 *	0.86
SOD (U/mL)	0.8 (0.4–1.1)	0.2 (0.1–0.6)	<0.01 *	0.4 (0.1–0.7)	0.4 (0.2–0.6)	0.96	0.01 *
Oxidants							
Lipid hydroperoxides _(_µmol)	3.3 (3.2–3.4)	3.3 (3.2–3.7)	0.69	3.3 (3.2–3.4)	3.4 (3.2–3.5)	0.66	0.57
Carbonyl groups in proteins (µmol)	3.0 (1.6–4.5)	4.0 (3.0–5.8)	0.05	3.6 (2.5–4.6)	3.6 (3.2–5.6)	0.55	0.15
8-Isoprostane (pg/mL)	36.4 (20.2–69.2)	19.2 (17.2–26.0)	0.01 *	24.2 (18.4–49.7)	28.4 (17.6–39.9)	0.91	0.37
Nitric oxide metabolites (µM)	323.7 (259.9–383.2)	249.6 (237.3–269.8)	0.01 *	270.2 (247.3–334.0)	255.5 (203.2–304.2)	0.29	0.09

Values are median (percentile 25–75), 8-OHG = 8-Hydroxy-2′-deoxyguanosine, hOGG1 = 8-Oxoguanine glycosylase, SOD = Superoxide dismutase, TNF-α = Tumor necrosis factor-alpha, IL-6 = Interleukin 6. U-MW = Mann-Whitney U test *p* < 0.05. * Comparison between overweight and obese hypertensive subjects.

## Data Availability

The database that supports the conclusions of this research work will be made available by the authors, upon express request and with the authorization of the Ethics and Research Committee.

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
