# Peer review of "Prevalence of Hypertension and Obesity: Profile of Mitochondrial Function and Markers of Inflammation and Oxidative Stress"

_antioxidants, 2023, doi:10.3390/antiox12010165_

Round 1
Reviewer 1 Report
Authors studies the potential implication of mitochondrial failure in plasma samples from hypertensive and obese t2dm patients. The work suggests interesting areas of research and connect mitochondrial damage with oxidative stress and inflammatory responses under the metabolic syndrome atmosphere. However, some issues must be cleared before being published.
- Mainly, why did you measure ATP in platelets and plasma from patients? What information are you obtaining and what tissue are you relating to? The myocardium will be one of the tissues involved, but not the only one and will not reflect specific responses of diabetic cardiomyopathy
- The age if obese patients is higher than lean subjects, and this can be a key factor for inflammatory responses. The number of hypertensive subjects is not convenient (n=32) and may not reflect the frequency of hypertension in this kind of populations. Also, in the obese cohort, only 16 of them were hypertensive.
- For t2dm criteria, did you evaluate the insulin resistance in patients?
- Some English type errors should be corrected
Author Response
Comment. Mainly, why did you measure ATP in platelets and plasma from patients? What information are you obtaining and what tissue are you relating to? The myocardium will be one of the tissues involved, but not the only one, and will not reflect specific responses of diabetic cardiomyopathy
Answer. Measurements related to ATP synthesis activity and ATPase activity were obtained from blood fractions. The results of this exploratory study reflect the state of mitochondrial function found in systemic blood and not in a specific organ or tissue. Peripheral blood analysis was sufficient to find differences in mitochondrial function in the studied population. Certainly, this measure has the limitation that it cannot attribute the main organs or tissues involved in these results. Determining the attribution in organs or other tissues was not contemplated within the objectives of the study.
Comment. The age of obese patients is higher than lean subjects, and this can be a key factor for inflammatory responses.
Answer. The prevalence of obesity has been rising steadily over the last several decades and is currently at unprecedented levels. This increase has occurred across every age, sex, race, and smoking status [[i]]
Comment. The number of hypertensive subjects is not convenient (n=32) and may not reflect the frequency of hypertension in this kind of population.
Answer. In the present study, only research subjects were included, and no patients were included. The subjects who participated in this study were volunteers who were not known to be ill. The subjects attended the check-up as healthy subjects without pathologies. In this sense, it was expected that the prevalence of hypertension would be different from that of the general population.
Hypertension displayed by the study subjects can be considered as undiagnosed or untreated hypertension. The present study shows a prevalence of undiagnosed hypertension of 18%, which is similar to that reported by other publications that contemplate a prevalence of approximately 13%. [[ii],[iii]]
Comment. Also, in the obese cohort, only 16 of them were hypertensive.
Answer. About the previous point, the subjects who participated in the study were volunteers who were known to be healthy. In this sense, the prevalence of hypertension found in this study may differ from those expected in the general population.
Comment. For the t2dm criteria, did you evaluate insulin resistance in patients?
Answer. For the diagnostic criteria of type 2 diabetes mellitus, only the fasting plasma glucose method was used. As suggested by the ADA, for the diagnosis of type 2 diabetes, fasting plasma glucose, 2-h plasma glucose during the 75-g oral glucose tolerance test, and A1C are equally appropriate [[iv]]
Comment. Some English-type errors should be corrected
Answer. I apologize. The grammar of the document was extensively revised, and we hope, it has improved significantly.
On behalf of the authors of the document, we appreciate your opinions and suggestions, which are intended to improve the writing and understanding of the study that we submitted to the Editorial Board of such a prestigious Journal.
Best regards
Alejandra Guillermina Miranda-Díaz, MD, PhD
[i]. Wright SM, Aronne LJ. Causes of obesity. Abdom Imaging. 2012;37(5):730-2
[ii]. Palomo-Piñón S, Antonio-Villa NE, García-Cortés LR, Álvarez-Aguilar C, González-Palomo E, Bertadillo-Mendoza OM, Figueroa-Suárez ME, Vargas-Hernández F, Herrera-Olvera IG, Cruz-Toledo JE, Cruz-Arce MA, Serafín-Méndez B, Muñoz-Cortés G, Morfin-Macias CJ. Prevalence and characterization of undiagnosed arterial hypertension in the eastern zone of Mexico. J Clin Hypertens (Greenwich). 2022;24(2):131-139
[iii]. Appleton SL, Neo C, Hill CL, Douglas KA, Adams RJ. Untreated hypertension: prevalence and patient factors and beliefs associated with under-treatment in a population sample. J Hum Hypertens. 2013;27(7):453-62
[iv]. American Diabetes Association. 2. Classification and Diagnosis of Diabetes: Standards of Medical Care in Diabetes-2021. Diabetes Care. 2021;44(Suppl 1):S15-S33
Reviewer 2 Report
After careful evaluation, I have to underline that the level of English throughout the manuscript does not allow easy reading and I strongly recommend consultation a professional language translating service.
Major comments:
1. The title says about „prevalence of hypertension and obesity with mitochondrial function..” whereas the authors assessed rather mitochondrial dysfunction in subjects with hypertension and obesity. In my opinion the title is misleading and should be changed.
2. In the study the authors aimed to „evaluate the prevalence of hypertension and obesity and the association of mitochondrial ATP synthase activity, inflammatory cytokines, and OS markers in subjects overweight or obese with or without hypertension. This statement should be rewritten.
3. There are several other statements to be rewritten such as : On page 1 in Introduction „Obesity is a chronic low-grade systemic inflammation” ; „Increased OS has been described in various atherosclerotic risk factors, such as hypertension”
4. What does the authors mean saying „No maintenance therapy was excluded”.
5. Discussion - lines 292 293
„This study evaluated the inflammatory profile, OS, and mitochondrial function in subjects without known pathological conditions. In the opinion of reviewer both obesity and hypertension mean pathology conditions.
Lines 325-327
„In the present study, the significant overexpression of the pro-inflammatory cytokines TNF-alpha and IL-6 in subjects with hypertension is striking, suggesting that hypertension and inflammation contribute to each other development”. This statement requires a basic and comprehensive explanation as it is well known over the years that inflammation is associated with hypertension and for example increased CRP which plays an important role in vascular inflammation is related with higher risk of hypertension occurrence.
Finally, the conclusions should also be rewritten, shortened and underline only the major most important findings.
Author Response
Comment. After careful evaluation, I have to underline that the level of English throughout the manuscript does not allow for easy reading and I strongly recommend consulting a professional language translating service.
Answer. I apologize, Paragraphs were spaced and some subheadings were included especially in the results. We hope that this will facilitate the reading and evaluation of the document that we offer for your consideration. The grammar of the document was extensively revised, and we hope, it has improved significantly.
Major comments:
Comment. The title says “Prevalence of hypertension and obesity with mitochondrial function.” whereas the authors assessed rather a mitochondrial dysfunction in subjects with hypertension and obesity. In my opinion, the title is misleading and should be changed.
Answer. We appreciate your comments and suggestions whose purpose is to improve the writing of the document and its relevance. We have taken into account your suggestion to change the title, however, with the modifications made to the original document, we are repurposing the title “Prevalence of Hypertension and Obesity: Profile of Mitochondrial Function and Markers of Inflammation and Oxidative Stress”. We will greatly appreciate your opinion on the matter.
Comment. In the study, the authors aimed to “evaluate the prevalence of hypertension and obesity and the association of mitochondrial ATP synthase activity, inflammatory cytokines, and OS markers in subjects overweight or obese with or without hypertension”. This statement should be rewritten.
Answer. To evaluate the prevalence of mitochondrial ATP synthase activity, inflammatory cytokines, and OS markers in subjects overweight or obese with or without hypertension
Comment. There are several other statements to be rewritten such as On page 1 in the Introduction ”obesity is a chronic low-grade systemic inflammation”; “Increased OS has been described in various atherosclerotic risk factors, such as hypertension”
Answer. The corrected text is the following: “Obesity is a systemic inflammation that promotes the immune system's activation, leading to a pro-inflammatory state and oxidative stress (OS)” and “OS and inflammation are key mechanisms of endothelial dysfunction and arterial damage, linking these risk factors to vascular disease and arterial stiffness” [[1]]
Comment. What do the authors mean by saying ”No maintenance therapy was excluded”
Answer. When interviewing to take the medical history of the subjects, some mentioned taking proton pump blockers for "gastritis" or analgesics for "headaches". Those drugs did not change
Comment. Discussion - lines 292 293, This study evaluated the inflammatory profile, OS, and mitochondrial function in subjects without known pathological conditions. In the opinion of the reviewer both obesity and hypertension mean pathology conditions.
Answer. The reviewer is right, the text was changed to “This study evaluated the inflammatory profile, OS markers, and mitochondrial function in subjects who did not know they were carriers of pathological conditions such as overweight, obesity, and Hypertension”.
Comment. Lines 325-327. „In the present study, the significant overexpression of the pro-inflammatory cytokines TNF-alpha and IL-6 in subjects with hypertension is striking, suggesting that hypertension and inflammation contribute to each other development”. This statement requires a basic and comprehensive explanation as it is well known over the years that inflammation is associated with hypertension for example increased CRP which plays an important role in vascular inflammation is related to higher risk of hypertension occurrence.
Answer. The reviewer is right, the association between hypertension and inflammation has been demonstrated; however, it is not clear whether inflammation is predominantly a cause or an effect of hypertension [[2]].
Comment. Finally, the conclusions should also be rewritten, and shortened and underline only the major most important findings.
Answer. Conclusions were modified accordingly
Dear reviewer 2.
On behalf of all the authors of the document, I would like to thank you for your opinions and suggestions, which are intended to improve the writing and understanding of the study that we made available to the Editorial Board of such a prestigious Journal.
Best regards
Alejandra Guillermina Miranda-Díaz, MD, PhD
[1]. Guzik TJ, Touyz RM. Oxidative Stress, Inflammation, and Vascular Aging in Hypertension. Hypertension. 2017;70(4):660-667
[2]. Dinh QN, Drummond GR, Sobey CG, Chrissobolis S. Roles of inflammation, oxidative stress, and vascular dysfunction in hypertension. Biomed Res Int. 2014;2014:406960
Round 2
Reviewer 2 Report
The manuscript has been sufficiently improved and all comments taken into account by the authors.